# Phenolic Composition, Antioxidant, and Anti-Proliferative Activities Against Human Colorectal Cancer Cells of Amazonian Fruits Copoazú (*Theobroma grandiflorum*) and Buriti (*Mauritia flexuosa*)

**DOI:** 10.3390/molecules30061250

**Published:** 2025-03-11

**Authors:** Sebastián Saldarriaga, Carlos Andrés Rodríguez-Salazar, Delia Piedad Recalde-Reyes, Gloria Magally Paladines Beltrán, Liceth N. Cuéllar Álvarez, Yudy Lorena Silva Ortíz

**Affiliations:** 1Grupo de Investigación en Productos Naturales Amazónicos-GIPRONAZ, Universidad de la Amazonia, Florencia 180001, Colombia; s.saldarriaga@udla.edu.co (S.S.); g.paladines@udla.edu.co (G.M.P.B.); li.cuellar@udla.edu.co (L.N.C.Á.); 2Molecular Biology and Virology Laboratory, Faculty of Medicine and Health Sciences, Corporación Universitaria Empresarial Alexander von Humboldt, Armenia 630008, Colombia; drecalde5552@cue.edu.co

**Keywords:** *Theobroma grandiflorum*, *Mauritia flexuosa*, antioxidant, antiproliferative

## Abstract

Amazonian fruits are a source of bioactive compounds, among which phenolic compounds, flavonoids, and carotenes stand out. These compounds play a crucial role in restoring oxidative balance, consequently reducing the proliferation of cancer cells. However, the content of these metabolites and their biological properties may vary significantly depending on the geographical location and the environmental conditions where plants grow. This research assessed the content of metabolites, free radical scavenging capacity, and hemolytic and antiproliferative effects of the hydro-methanolic extracts of the Amazonian fruits *Theobroma grandiflorum* and *Mauritia flexuosa*. The results revealed that the extracts derived from the seeds of *Theobroma grandiflorum* sourced from the Balcanes experimental farm and the pulp of *Mauritia flexuosa* harvested in Florencia exhibited higher contents compared to other analyzed sites: Total phenolic content (TPC) (619.41 ± 12.05 and 285.75 ± 10.06 mg GAE/100 g FW), Total flavonoid content (TFC) (569.09 ± 4.51 and 223.21 ± 3.92 mg CAT/100 g FW), and Total carotenoid content (TCC) (25.12 ± 0.16 and 48.00 ± 0.28 mg eq β-carotene/100 g FW), respectively. Also, these samples demonstrated superior scavenging capacities for the ABTS and DPPH radicals, while the peel of *Mauritia flexuosa* exhibited the highest scavenging capacity for the oxygen radical (526.23 ± 2.08 µmol Trolox.g^−1^). The hemolytic effect shows dose-dependent responses with IC50 values of 27.73 μg/mL for the Balcanes seeds and 1.27 μg/mL for the Florencia pulp. Furthermore, it was observed that treatment with the fruit-derived extracts effectively reduced the number of viable human colorectal cancer cells, using SW480 ATCC cell line, demonstrating a non-dose-dependent behavior compared to the control cells.

## 1. Introduction

Cancer is considered an abnormal cellular event involving the loss of control over cell growth and division, mediated by genetic and epigenetic events [1]. One of the primary causes of cancer development is associated with oxidative stress, characterized by a disproportionate increase in reactive oxygen and nitrogen species (ROS and RNS) that disrupt the balance between pro-oxidant agents and the endogenous antioxidant system at the cellular level [2] this can lead to the degradation of primary biomolecules such as DNA, where their alteration could be a potential cause of degenerative, inflammatory, and proliferative conditions, intrinsically associated with cancer [3,4].

Lifestyle and dietary habits are also among the environmental factors linked to carcinogenesis, assuming prominence as potential initial triggers for cellular imbalance scenarios [3]. An illustrative example of this correlation is observed in diets characterized by low intake of natural antioxidants, such as phenolic compounds like flavonoids, anthocyanins, and other liposoluble compounds found in fruits and vegetables. This dietary pattern correlates with a heightened probability of cancer development, particularly malignancies associated with the digestive system, such as colorectal cancer [5,6,7].

In Colombia, colorectal cancer ranks as the third most common cancer among both men and women. The risk of mortality is notably high in the country’s major cities and is closely associated with dietary habits adopted by the population. Specifically, there is a prevalent pattern of high consumption of ultra-processed foods and low intake of fruits and vegetables, averaging 45 kg per year, which is significantly lower than the 120 kg per year recommended by the World Health Organization (WHO) for a healthy and functional diet [8]. This starkly contrasts the country’s abundant variety of traditionally cultivated fruits and vegetables and the vast array of Amazonian fruits thriving in the Colombian Amazon. Amazonian fruits, besides being recognized as a rich source of macronutrients such as vitamins and minerals [9], exhibit a wide array of primary and secondary metabolites, including phenolic derivatives, flavonoids, anthocyanins, phytosterols, tocopherols, carotenoids, sugars, among others [10,11]. These compounds play a crucial role in restoring oxidative balance disrupted by free radicals [8]. Among the Amazonian fruits, Copoazú (*Theobroma grandiflorum Willd. ex Spreng. K.*) and Butiri (*Mauritia flexuosa Lf.*), belonging to the Malvaceae and Arecaceae families respectively, stand out. Both are widely distributed throughout the Amazon River basin, especially in southern Colombia and northwestern Brazil [12].

The pulp of *Theobroma grandiflorum* (*T. grandiflorum*) possesses a strong aroma, creamy texture, pleasant flavor, and high acidity, and it is commonly consumed in juices, beverages, ice creams, jellies, and sweets. Meanwhile, the seeds contain significant amounts of fatty acids, which are valuable in the food industry for the preparation of ‘Copulate’ and in cosmetic formulations [13]. Research has shown that the pulp can reduce lipid peroxidation and restore antioxidant capacity in rodents on cholesterol-rich diets. Additionally, fermented seed extracts protect the heart against ischemia-reperfusion damage through nitric oxide synthase (NOS)-dependent mechanisms [14,15]. Therefore, in traditional medicine, ziora-anema from the Colombian Amazon has been positioned as a non-pharmacological adjuvant tool in reducing diabetic complications triggered by nitrosative stress and inflammation in the kidneys [16].

Similarly, the *Mauritia flexuosa* (*M. flexuosa*) fruit is a significant source of bioactive compounds and beneficial fatty acids for health. These compounds are capable of aiding in the prevention of oxidative stress and chronic and degenerative diseases [17]. The pulp is orange-colored, smooth, and water-soluble, with a significant oily fraction rich in carotenoids, tocopherols, ascorbic acid, phenolic compounds, and monounsaturated fatty acids, primarily represented by oleic acid. The extraction and commercialization of oil from this palm tree are considered the main form of utilization by Amazonian communities [18,19].

To date, several studies have demonstrated how phenolic compounds, their derivatives, and multiple secondary metabolites, such as carotenoids, associated with Amazonian fruits, exhibit a prominent antioxidant effect and effectively reduce the proliferation of cancer cells in vitro through various mechanisms, including redox restoration or activation of cellular senescence pathways [20,21]. Based on the above information, this research aimed to evaluate the content of phenolic compounds, antioxidant and antiproliferative activity against human colorectal cancer cells of Amazonian fruits *T. grandiflorum* and *M. flexuosa*.

## 2. Results

### 2.1. Phenolic Compounds of T. grandiflorum and M. flexuosa

The total content of phenols, flavonoids, and carotenoids in the evaluated Amazonian fruits is depicted in Table 1. The seeds of *T. grandiflorum* and the pulp of *M. flexuosa* exhibited elevated levels of these compounds compared to their respective pulp and peel. Furthermore, the content of these secondary metabolites varied significantly depending on the sampling site (*p* < 0.05). For *T. grandiflorum*, seeds of Balcanes origin showed the highest values for total phenolic content (TPC): 619.41 ± 12.05; TFC: 569.09 ± 4.61; TCC: 25.12 ± 0.16; Gallic acid: 335.12 ± 10.87; Catechin: 39.86 ± 0.29; Quercetin: 70.11 ± 0.48; and, for the *M. flexuosa* fruit pulp, obtained from the Amazon Research Center Macagual (CIMAZ) exhibited superior phenolic content compared to other sources (TPC: 508.48 ± 10.06). Conversely, the total flavonoid content (TFC), total carotenoid content (TCC), and phenolic compounds determined by HPLC-UV were higher in pulp of Florencia origin (TFC: 223.21 ± 3.92; TCC: 48.00 ± 0.28; Gallic acid: 219.54 ± 0.68; Catechin: 61.15 ± 0.56; Quercetin: 95.50 ± 2.80) (Table 1).

### 2.2. Antioxidant Activity of T. grandiflorum and M. flexuosa

Total antioxidant activity, assessed through oxygen radical absorbance capacity (ORAC), revealed that both *T. grandiflorum* seeds and *M. flexuosa* peel showed remarkable antioxidant activity, ranging from 50.22 ± 0.41 to 526.23 ± 2.08. Specifically, samples of *T. grandiflorum* seeds from Balcanes and *M. flexuosa* peel from Montañita presented the highest antioxidant activity, reaching 204.93 ± 0.41 and 526.23 ± 2.08, respectively (Table 2).

On the other hand, the radical scavenging capacity of ABTS ([2,2-azino-bis(3-ethylbenzothiazoline-6-sulfonate]) and DPPH (2,2-diphenyl-1-picrylhydrazyl) radicals varied depending on the portion of the fruit and the sampling site evaluated. *T. grandiflorum* seeds from the Balcanes experimental farm exhibited the highest radical scavenging capacity (207.63 ± 1.64 ABTS, 117 ± 1.24 DPPH), while for *M. flexuosa*, the pulp from Florencia showed better indices (71.99 ± 0.65 ABTS and 67.23 ± 0.59 DPPH).

### 2.3. Principal Component Analysis

The principal component analysis (PCA) reveals variables associated with secondary metabolite content and their radical scavenging capacity, transforming them into a new set of uncorrelated variables that synthesize the most significant variability associated with the data. For *T. grandiflorum* (Figure 1A), PCA elucidated over 90.9% of the observed variability in the first two components (PC1: 0.755 and PC2: 0.154), effectively distinguishing between pulp (P) and seeds (S) based on the sampling site (B: Balcanes, C: Caraño, V: Versalles) (*p* < 0.05). Notably, strong correlations were found between TPC: 0.97, TFC: 0.98, TCC: 0.91), DPPH: 0.97, and ORAC: 0.97 with the first component (PC1). Conversely, variables reflecting specific phenolic derivatives established by HPLC-UV techniques were associated with the second component (PC2), including Catechin (0.75), Quercetin (0.43), and Gallic acid (0.57).

The PCA also revealed that the antioxidant activity assessed by ORAC is significantly influenced (*p* < 0.05) by the TPC: 0.93, TFC: 0.97, and TCC (0.95). Furthermore, this variable was strongly correlated with the DPPH radical scavenging capacity (0.99). Regarding the ABTS and DPPH radical scavenging capacity, a significant correlation was observed with variables such as TFC, TCC (>0.9), and TFC (>0.8). Seeds from Balcanes (B), Caraño (C), and Versalles (V) origins reached the maximum values and exhibited the strongest relationship with the variables in this study. For that reason, seed extracts were selected for hemolytic and cell viability analyses.

The PCA also elucidated around 94% of the variability associated with the data in the first two components (PC1: 0.793 and PC2: 0.147) for *M. flexuosa* fruit (Figure 1B), discriminating between peel (C) and pulp (P), across different sampling sites (Mon: Montañita, Mac: Macagual and Flo: Florencia). Notably, the pulp from Florencia and Montañita origins exhibited higher values in the analyzed variables. Furthermore, variables such as TCC (0.99), TFC (0.95), Quercetin (0.99), DPPH (0.97), ABTS (0.98), and Catechin (0.99) were correlated with the first component (PC1), while the TPC and gallic acid associated with the second component (0.93 and −0.57). The ABTS and DPPH radical scavenging capacity for *M. flexuosa* fruit was primarily influenced by the TCC: ABTS = 1, DPPH = 0.97, TFC: ABTS = 0.90, DPPH = 0.98, as well as specific flavonoids like catechin: ABTS = 0.96, DPPH = 0.98 and quercetin: ABTS = 0.94, DPPH = 0.96). Additionally, ORAC exhibited a negative correlation with the analyzed variables, suggesting that none of the quantified metabolites in this study explain or contribute to this antioxidant behavior. However, the peel of Florencia origin displayed the highest value (Figure 1B).

### 2.4. Antiproliferative Activity of Extracts T. grandiflorum and M. flexuose

Ex-Vivo human red blood cell hemolysis assay was performed to assess the safety of the extracts and their potential human antitumoral effects tested on SW480 ATCC cells, a human cancer cell line derived from a primary adenocarcinoma of the colon compared. We found that the Peel of *M. flexuosa* from Florencia did not exhibit a hemolytic effect. In contrast, the Pulp of *M. flexuosa* from montañita exhibited a hemolytic effect only at 100 µg/mL, other extracts showed hemolytic effects at concentrations around 12.5 µg/mL. The morphology of erythrocytes was observed by light microscopy (40×, Zeiss^TM^ Primo Star, Oberkochen, Germany) at the end of the assay, revealing no shape alterations at low concentrations (0.78–6.25 µg/mL), where no hemolysis was detected (Figure 2A–G).

According to Figure 3, the extracts obtained from the peel of *M. flexuosa* from Florencia, the seeds of *T. grandiflora* from Versalles, the seeds of *T. grandiflora* from Balcanes, and the seeds of *T. grandiflora* from Caraño exhibited antitumoral effects when tested on SW480 ATCC cells. These extracts reduced cell viability by approximately 50% at low concentrations. Interestingly, all extracts reduced the viability of SW480 ATCC cell lines, contrasting with the hemolysis assay results where concentrations of 0.78–6.25 µg/mL did not affect the erythrocytes.

## 3. Discussion

The Amazonian rainforest of Caquetá, Colombia, is a prosperous region in biodiversity. It also houses a wide variety of unique plant species and many Amazonian fruits. This region is in the Andean Amazonian area and has been traditionally used by indigenous communities for agricultural purposes [22,23]. The geography and topography of Caquetá, Colombia contribute to its unique micro-climates, making it an ideal environment to produce several fruits such as acai berries (*Euterpe oleracea*), Camu Camu (*Myrciaria dubia*), and Copoazú (*T. grandiflorum*) and Butiri (*M. flexuosa*). These fruits are essential for the local communities’ diet and have gained attention for their potential health benefits and commercial value [14,24,25].

The diverse range of fruits in Caquetá presents an opportunity for further research into their nutritional content, medicinal properties, and economic potential. The nutritional value, bioactive compounds, and potential uses of these Amazonian fruits shed light on their significance for local consumption and international trade [14,22,24,25,26]. Therefore, in this paper, we focus on testing secondary metabolites associated with Amazonian fruits *T. grandiflorum* and *M. flexuosa*; the total content of phenols, flavonoids, and carotenoids in these Amazonian fruits where were exhibited elevated levels of these compounds compared to their respective pulp-peel or pulp-seed, however the content varied depending on the sampling site, for *T. grandiflorum*, seed of Balcanes origin showed the best values for total phenolic content (TPC), while for *M. flexuosa* were for fruit pulp, from Macagual. Conversely, the total flavonoid content (TFC), total carotenoid content (TCC), and phenolic compounds were higher in the pulp of Florencia origin (Table 1). Pugliese et al. [27] reported *T. grandiflorum* phenolic proportions between 4–5 times higher in seeds than fresh pulp when using highly polar solvents such as methanol and water in a 7:3 ratio, like the present study. This difference is mainly associated with the strong content of flavan-3-ol type phenolic derivatives reported in the seeds, among which stand out glucuronide-8-O-β-D-hipolaetin, glucuronide 3″-O-sulfate-8-O-β-D-hipolaetin (theo-grandina II), glucuronide 8-O-β-D-isoscutelarin, glucuronide-3′-methylether-8-O-β-D-hipolaetin, glucuronide-3″-O-sulfate-8-O-β-D-isoscutelarin (theograndin I), and glucuronide-3′-methylether-″-O-sulfate-8-O-β-D-3-hipolaetin [27]. These metabolites are highlighted for their role as protective agents against pathogens as a defense mechanism against predators, plant competition, and abiotic stress [28].

On the other hand, for *M. flexuosa*, we found that the concentration of these metabolites in the peel and p–ulp varied significantly (*p*-value 0.0001). This difference could be attributed to phenolic derivatives in the pulp. Authors such as Abreu-Naranjo et al. [29] highlight the presence of significant flavonoids such as quercetin-dihexoside, myricetin glucuronide, methyl myricetin-O-glucuronide, quercetin-O-rutinoside, quercetin-O-glucoside, quercetin-3-O-glucuronide, kaempferol-3-O-glucoside, kaempferol-3-O-glucuronide, naringenin hexoside, luteolin-O-deoxyhexoside, quercetin, cyanidin-3-rutinoside, and cyanidin-3-glucoside, which serve important physiological functions in the plant. This contrasts with the peel, where lignin and cellulose predominate.

The fruits of *T. grandiflorum* from the Balcanes experimental farm, located 250 m above sea level, displayed markedly higher levels of bioactive compound content in the seeds. On the other hand, the locality of Caraño, located at 750 m above sea level, lies in a transitional zone between the Colombian Andes and the Amazon. Its environment encompasses a variety of ecosystems, including very humid tropical forests, very humid premontane forests, and humid montane forests, characterized by rolling hills, high cloud cover, and continuous leaching of nutrients and organic matter [29]. This highlights that cultivation areas with moderate humidity, vegetation thickness, and ample luminosity, such as the Balcanes experimental farm, favor the low degradation of the organic matter cycle due to climatic factors, as the bioactive compounds analyzed are synthesized from carbon skeletons of primary metabolism obtained from the soil organic matter cycle by the plant [29]. For the *M. flexuosa* fruit, the total phenols, flavonoids, and carotenoids also varied significantly depending on the harvest location; for example, the Florencia site showed the highest contents. This variation could be attributed to environmental and edaphological conditions associated with the origin. Factors such as climate, solar radiation, soil mineral richness, plant development-related factors, and the distribution of secondary metabolites in the metabolic synthesis of their derivatives are among the aspects of chemical ecology that can affect and limit their content [30].

The environmental conditions surrounding the cultivation of *T. grandiflorum* seeds and *M. flexuosa* peel play a crucial role in determining their antioxidant activity. As previously discussed, the fruits of T. grandiflorum from the Balcanes experimental farm exhibited significantly higher levels of bioactive compounds in the seeds compared to other locations, highlighting the influence of geographical and environmental factors on plant metabolism. Similarly, the ecological characteristics of the Florencia site contributed to the remarkable antioxidant activity observed in the *M. flexuosa* peel. Total antioxidant activity, evaluated through oxygen radical absorbance capacity (ORAC), revealed noteworthy results for both *T. grandiflorum* seeds and *M. flexuosa* peel. Specifically, samples from Balcanes and Florencia showed the highest antioxidant activity. It is worth noting that while these values are lower than those reported by Tauchen et al. [31] for *T. grandiflorum* pulp and *M. flexuosa* peel and pulp in the Peruvian Amazon, they still demonstrate significant antioxidant potential.

Phenolic derivatives are characterized by low oxidation potentials, allowing them to be oxidized before other chemical species and consequently exert a protective effect against oxidative attack (free radicals, ROS, RNS, light, etc.), acting as antioxidants or radical scavengers due to the surplus π system present in their aromatic structure, which provides the ability to donate electrons and restore their reduced form through the conjugated system of double bonds. However, complex structures like flavonoids can return to their reduced form through the interaction of other functional groups present in their different rings, whereby once oxidized, they can regain their hydroxyl group along with their antioxidant capacity, preventing the oxidation of other elements of interest [32]. In the case of carotenoids, their structure is considered an electron-rich system susceptible to attack by electrophiles, allowing them to be oxidized first and consequently exert a protective effect against oxidative attack. β-carotene has a more significant number of unsaturations than α-carotene and γ-carotene, thus increasing its ability to capture electrophilic species with the number of conjugated unsaturations. In this way, they neutralize reactive species such as singlet oxygen and free radicals like the hydroxyl radical (HO·) and peroxide radicals, forming the basis of their antioxidant action in biological systems [33].

On the other hand, the literature indicates that 50 mg/L of optimized lyophilized camu camu extract (*Myciaria dubia*) achieves 50% erythrocyte lysis [34]. However, substances are known to cause less than 1% hemolysis, such as anthocyanin-enriched extracts from *Cichorium intybus* or the novel water-soluble polysaccharide (SWSP) extracted from *Sorghum bicolor* (L.), another plant commonly used as food [35,36].

Other cell lines have observed this non-dose-dependent behavior at specific evaluated concentrations. Authors such as Villada et al. [37] assessed the impact of *Passiflora edulis* ethanolic extract on cell viability of SW480 ATCC and HFF (non-tumoral fibroblast cells) lines. They found that the concentration of 500 µg/mL induced a more significant decrease in cell viability compared to concentrations of 1000, 1500, 2000, and 3000 µg/mL 24 h after treatment, possibly due to the synergy of bioactive compounds in the leaf extract. However, the authors reported an IC50 of 1000 μg/mL (81.1%) at 24 h without disclosing the R^2^ value of their model. Dahabiyeh et al., [38] investigated the effect of 7 analogs of the 2,3-dihydroquinazolin-4(1H)-one structure on cell viability in HTC-116 and SW480 ATCC lines. They found that the derivative compound 5-(2-(4-(Dimethylamino)phenyl)-2,3-dihydroquinazolin-4(1H)-one) at 12.5 μM generated a lower percentage of viable cells compared to concentrations of 25 and 50 μM in the SW480 ATCC cell line. A similar trend was observed for the compound 3-(2-(4-chlorophenyl)-2,3-dihydroquinazolin-4(1H)-one) at concentrations of 25 and 50 μM in the HCT116 line, suggesting that higher concentrations do not necessarily imply a more significant impact on cell viability.

While this is the first approach to the potential use of these extracts in anticancer applications, these results suggest that the extracts from these plants could have anticancer effects. Therefore, further studies are needed to investigate their impact on cancer cells, such as conducting analyses of these extracts directly on cancer patient biopsies. Additionally, exploring other biological effects, such as antiparasitic, antiviral, or antibacterial properties, would be beneficial.

## 4. Materials and Methods

### 4.1. Reagents

Methanol, Folin-Ciocalteu reagent (PanReac/Applichem, Darmstadt, Germany), Anhydrous sodium carbonate (Pan-Reac/Applichem, Darmstadt Germany), Gallic acid (Alfa Aesar, Morecambe, UK), Aluminum chloride (Alfa Aesar, Morecambe, UK), Sodium nitrite (PanReac/Applichem, Darmstadt, Germany), Sodium hydroxide (PanReac/Applichem, Darmstadt, Germany), (+)-Catechin, Acetone (PanReac/Applichem, Darmstadt, Germany), Phenol (PanReac/Applichem, Darmstadt, Germany), Sulfuric acid (PanReac/Applichem, Darmstadt, Germany), D-glucose (PanReac/Applichem, Darmstadt, Germany), Dinitrosalicylic acid (PanReac/Applichem, Darmstadt, Germany), ABTS (Merck, Rahway, NJ, USA), Potassium persulfate (Sigma Aldrich, St. Louis, MO, USA), Trolox (Sigma Aldrich, St. Louis, MO, USA), DPPH (Sigma Aldrich, St. Louis, MO, USA), DMEM medium (Capricorn Scientific, Ebsdorfergrund, Germany), Methanol (Merck, Rahway, NJ, USA), Ethanol (Merck, Rahway, NJ, USA), Penicillin/streptomycin antibiotic 10× (Sigma Aldrich, St. Louis, MO, USA), Trypsin-EDTA (Sigma Aldrich, St. Louis, MO, USA), NaCl (Merck, Rahway, NJ, USA).

### 4.2. Sampling and Fruit Collection Area

The fruits of *T. grandiflorum* were collected in three locations in southern Colombia in the Amazon region (Caquetá). The climate in Caquetá is tropical, with an average relative humidity of 84%, a solar radiation of 1465.4 h of light, and approximately 3669 mm of precipitation per year [39]. The first location was the Balcanes experimental farm of the University of the Amazon, situated in Florencia, Caquetá, 250 m above sea level (1°37′04″ N and 75°36′04″ W). This area is characterized as a tropical humid forest, with an average annual precipitation of 3793 mm, an average annual temperature of 25 °C, and an average annual relative humidity of 84% [40]. The second location was the Caraño area, located along a mountain range with an altitudinal gradient ranging from 700 to 2200 m (1°43′54″ N and 75°38′24″ N), encompassing very humid tropical forest, very humid premontane forest, and humid montane forest [29]. The third location was the Versalles area, located south of the department, at 257 m above sea level (1°12′04″ N and 75°56′04″ W).

For the *M. flexuosa* fruit, sampling sites were selected in the city of Florencia (1°35′20″ N and 75°37′39″ W), with an average annual precipitation ranging between 3500 and 3700 mm, an average annual temperature of 24.5 °C, and a zone classified as Tropical Humid Forest within the Colombian Amazon. The site La Montañita (1°29′51″ N and 75°24′13″ W), with a mean temperature of 25.5 °C and annual precipitation of 3793 mm, is classified as a tropical rainforest according to Olaya [41]. Finally, the Macagual research center (1°29′45″ N and 75°24′31″ W) is located in a humid area with an average annual precipitation of 3793 mm (rainforest), an annual light intensity of 1707 h, an average temperature of 25.5 °C, and a relative humidity of 84.3% [42].

The Caraño trail is located along a derivation of the mountain range, which presents an altitudinal gradient of 700–2200 m (1°43′54″ N and 75°38′24″ N), including very humid tropical forest, very humid premontane forest and lower montane humid forest [26] and the Versalles trail, located to the south of the department at 257 m.a.s.l. (1°12′04″ N Y 75°56′04″ W). In the case of the *M. flexuosa* fruit, the municipality of Florencia (1°35′20″ N and 75°37′39″ W) was selected as the sampling point with average annual precipitation between 3500 and 3700 mm, average annual temperature of 24.5 °C and Tropical Humid Forest area in the Colombian Amazon, the municipality of La Montañita (1°29′51″ N Y 75°24′13″ W) with an average temperature of 25.5 °C, annual rainfall of 3793 mm and classified as tropical rainforest according to Olaya [41]; and the Macagual research center, located in a humid area, with average annual precipitation of 3793 mm (rain forest), light intensity per year of 1707 h, average temperature 25.5 °C and relative humidity of 84.3% (1°29′45″ N and 75°24′31″ W) [42]. During the post-harvest, the fruits were transported to the laboratory under refrigeration. Manual extraction was performed to separate the pulp and peel (*M. flexuosa*) and pulp and seeds (*T. grandiflorum*).

### 4.3. Sample Preparation

6.0 g of the fruit of *T. grandiflorum* (pulp and seed) and *M. flexuosa* (peel and pulp) were taken, 30 mL of methanol: water (7:3 *v*/*v*) was added, and it was shaken vigorously for 30 min and were ultrasonic (frequency 40 KHz) using sonication equipment (Best Built Jewelry Equipment 28–330, Seoul, Republic of Korea) for 15 min with distilled water at 25 °C. It was then centrifuged in (SL 8R Thermo Fisher Scientific, St. Louis, MO, USA) at 4500 rpm for 10 min at 4 °C, and the supernatant was filtered [27]. The resulting extract was divided into two fractions. The first 10 mL was used to analyze secondary metabolites and antioxidant capacity in the MultiSkan Go plate reader (Thermo Fisher Scientific). The remaining 20 mL were used for hemolytic and viability assays on SW480 ATCC cells. For this purpose, the solvent was removed using a RE100-Pro rotary evaporator (DLAB, Beijing, China). The first 10 mL was used to analyze secondary metabolites and antioxidant capacity in thetor (DLAB, China) at 35 °C until a volume of 2 mL was reached. It was preserved at −40 °C for subsequent NBJ lyophilization (NANBEI, Beijing, China). The samples were stored at 4 °C, and all assays were performed using triplicate.

### 4.4. Determination of Bioactive Compounds

#### 4.4.1. Total Phenolic Content (TPC)

The total phenolic content was determined by spectrophotometry using the Folin-Ciocalteu colorimetric method [43]. To do this, 18 µL of the extract, 124.5 µL of deionized water, 37.5 µL of Folin-Ciocalteu reagent, and 120 µL of anhydrous sodium carbonate (Na_2_CO_3_) at 7.1% were taken. The reaction was allowed for 60 min in the dark at room temperature, then the absorbance was read at 760 nm. Gallic acid was used as reference standard for quantification purposes in a concentration range from (10–550 ppm, R^2^: 0.9986). The results were expressed as mg of gallic acid (GA)/100 g of fresh fruit [44].

#### 4.4.2. Total Flavonoid Content (TFC)

The total flavonoid content was determined using a modified version of the method developed by Woisky et al. [45], which involves a reaction with aluminum chloride (AlCl_3_). Briefly, the reaction mixture consisted of 120 µL of deionized water, 30 µL of the extract, 9 µL of 5% sodium nitrite (NaNO_2_) (left for 5 min), 9 µL of 10% aluminum chloride (AlCl_3_) was added, and left for 5 min, then 60 µL of 1 M sodium hydroxide (NaOH) was added, allowing it to react for 15 min and, finally, 72 µL of deionized water were added. The reaction was carried out in the dark at room temperature for 30 min, and the absorbance was read at 510 nm. (+)-catechin (10–550 ppm, R^2^: 0.9928) was used as a standard pattern; the results were expressed in mg of catechin (CT)/100 g of fresh fruit.

#### 4.4.3. Total Carotenoid Content (TCC)

The carotenoid content was quantified according to the method proposed by Neves et al. [46] with slight modifications. 2 g of fruit sample was used, 10 mL of cold 99.5% (*v*/*v*) acetone was added, it was shaken vigorously for 30 min in LP Vortex Mixer (Thermo Scientific, Seoul, Republic of Korea) at 3000 rpm, then it was taken to ultrasound (frequency of 40 KHz) using equipment (Best Built Jewelry Equipment 28-330, Republic of Korea) for 15 min with distilled water at 25 °C, then centrifuged in (SL 8R Thermo Fisher Scientific, Branchburg, NJ, USA) at 4 °C and 4500 rpm for 10 min, finally, the supernatant was filtered. The absorbance of the extract was read in a quartz cell at 450 nm. As a standard, β-carotene (10–550 ppm, R^2^: 0.9974) was used, and the results were expressed as mg of β-Cat/100 g of fresh fruit.

### 4.5. Quantification of Phenolic Compounds by (HPLC-UV)

100 mg of each sample was homogenized in 1 mL of methanol: water: formic acid (25:24:1, *v*/*v*/*v*) for 60 min with Best Built Jewelry Equipment 28-330, Republic of Korea), stored at 4 °C throughout the dark. Then, each extract was centrifuged at 10,000× *g* for 15 min, and the supernatants were passed through PVDF syringe filters (0.22 µm, mmØ, Analysis Vinicos, Tomelloso, Spain).

Phenolic compounds were quantified utilizing an HPLC-UV system (Shimadzu LC-2010A HT) equipped with a Luna C18 column (250 mm × 4.6 mm, 5 µm; Luna Phenomenex, Macclesfield, UK). The injection volume was 30 µL, with chromatograms recorded at 280 nm and 320 nm wavelengths. The phenolic compounds were identified based on their UV-Vis spectra and retention times (tR). The phenolic compounds were quantified using gallic acid as standard 280 nm and flavonoids using catechin and quercetin as standards at 280 nm. The mobile phase was composed of the solvent (A) water/formic acid (99:1, *v*/*v*) and (B) acetonitrile. The flow was 0.8 mL/ min in a linear gradient, starting with 6% B for 5 min, reaching 24% B at 18 min, maintained for 7 min, then 60% B at 30 min, and finally to 95% B at 40 min. All samples were extracted in triplicate and injected three times. All standard curves were run in the range of 0.015–0.031 mM up to 1 mM (stock solution), the LOQ (limit of quantification) was close to 0.01 mM in each case, and the R^2^ was more significant than 0.99. The results were expressed as mg equivalent of the standard/100 g of dry weight.

### 4.6. Antioxidant Activity

#### 4.6.1. Determination of Total Antioxidant Capacity by Oxygen Radical Absorption (ORAC)

A 0.04 μM fluorescein solution was prepared in phosphate buffer (0.075 M, pH = 7.0 (2600 μL), mixed with 100 μL of sample in a 4 mL test tube. The reaction mixture was incubated for 30 min at 37 °C, and the antioxidant capacity was determined by observing the decay of the initial fluorescence by adding a radical generator, as Ou et al., described [47]. For this reason, 300 μL of 200 mM AAPH solution prepared in phosphate buffer (0.075 M, pH = 7.0) was added to the above mixture, and after stirring, it was deposited in glass cuvettes (1 × 1 × 3.5 cm). Fluorescence measurements were performed at 37 °C using an RF-6000 spectrofluorometer (Shimadzu, Kyoto, Japan). The excitation and emission wavelengths were 485 nm and 520 nm, respectively. Fluorescence was monitored continuously for 60 min. The 0.04 μM fluorescein solution was prepared in 0.075 M phosphate buffer, pH = 7.0, with 100 μL 0.075 M phosphate buffer, pH = 7.0, and was used as a sample blank.

The antioxidant capacity, expressed as AUC, was calculated as:*AUC* = 1 + *f*1/*f*0 + …*fn* + 1/0
where:AUC: area under the fluorescence decay curvef0: the initial fluorescence reading at 0 minfn: the fluorescence reading at time n.

The solution concentration of the examined extracts and trolox were equalized to 0.1 mM.

#### 4.6.2. ABTS Assay

Antioxidant activity was determined according to the method of Neves et al. [46], with some modifications. As a first measure, the radical cation ABTS+ was generated; for this, the ABTS stock solution (10 M) was oxidized with potassium persulfate (2.45 M). The resulting mixture was allowed to stand in the dark at room temperature for 16 h before use. The ABTS·+ radical solution was diluted in 0.15 M saline phosphate buffer pH 7.4 until obtaining an absorbance of 0.7 units. For the test, 3 µL of the extract and 297 µL of the radical solution were used. It was allowed to react for 30 min in the dark, and the absorbance was read at 734 nm wavelength, considering the radical reference blank. The results were expressed as TEAC values in µmol of Trolox/g of fresh fruit, using a Trolox standard curve (10–650 ppm, R^2^: −0.9977) [44].

#### 4.6.3. DPPH Radical Scavenging Capacity

The DPPH (1,1-diphenyl-2-picrylhydrazyl, Sigma-Aldrich, Burlington, MA, USA) radical scavenging activity was determined using a method adapted from Suarez et al. [44] with slight modifications; A stock solution of DPPH (20 mg/L) was prepared in 99% methanol at 4 °C. The radical absorbance was adjusted to 0.3 units with 99% methanol at 4 °C. For the reaction, 3 µL of the extract and 297 µL of the DPPH stock solution were taken and allowed to react in the dark for 30 min at room temperature, and the absorbance was read at a wavelength of 517 nm. The results were expressed as TEAC values in µmol of Trolox/g of fresh fruit, using Trolox as a standard (0.1–1 ppm, R^2^: −0.9953) [46].

### 4.7. Antiproliferative Activity

#### 4.7.1. Hemolytic Activity

4 mL of A+ blood was collected using EDTA as an anticoagulant and centrifuged at 345× *g* for 5 min. The plasma was removed, and the erythrocytes were washed three times with saline solution (0.9%). Subsequently, 200 µL of the erythrocytes were taken and diluted in a 1:20 ratio with saline solution, then incubated at 37 °C in a 5% CO_2_ atmosphere for 20 min. A second dilution of 1:5 was performed from the previously prepared dilution for the test. The erythrocytes were exposed to each sample through a two-fold serial dilution starting from 100 µM to 0.78 µM (covering both fruit and pulp) for 30 min at 37 °C in a 5% CO_2_ atmosphere. After this period, the plate was centrifuged at 800× *g*, the supernatants were collected, and the absorbance was measured at 540 nm using a microplate reader (Epoch Biotek^®^, Winooski, VT, USA) [48].

#### 4.7.2. Cell Culture

Human colorectal cancer cells (SW480 ATCC) were obtained from the American Type Culture Collection (Manassas, VA, USA) and maintained in Dulbecco’s Modified Eagle’s Medium (Gibco by Life Technologies, Rockville, MD, USA) supplemented with 10% fetal bovine serum (Life Technologies, Rockville, MD, USA) and 100 IU/mL penicillin-streptomycin (Eurobio Scientific, Les Ulis, France). The cells were cultured in a 5% CO_2_ atmosphere at 37 °C and grown to 80% confluency before treatment with fruit extracts. The compounds under study were dissolved in culture medium for all experiments and incubated for 24 h at 37 °C in a 5% CO_2_ atmosphere.

#### 4.7.3. Determination of Cell Viability by Incorporating Crystal Violet

To determine the number of viable cells post-treatment, the inoculum was first wholly removed from the plates. The cells were washed with 1X PBS (Sigma-Aldrich, St. Louis, MO, USA) and fixed with 4% formaldehyde (Sigma-Aldrich, St. Louis, MO, USA) in 1X PBS for 15 min. After fixation, the cells were rewashed with 1X PBS to remove residual fixative. Subsequently, the cells were stained with 0.1% crystal violet dye (Sigma-Aldrich, St. Louis, MO, USA) in a PBS buffer solution adjusted to pH 6 for 20 min. To remove excess dye, the cells were washed with 1X PBS. Finally, the dye was solubilized using 10% ethanol at 200 µL per well (J.T. Baker, Deventer, Holland), and the absorbance was measured at 570 nm using a spectrophotometer (Biotek^®^, Winooski, VT, USA)

### 4.8. Statistical Analysis

Each experimental determination was conducted in triplicate, and the values presented are the mean ± SD from three independent experiments. A generalized linear and mixed model (GLMM) analysis was utilized to compare the response variables. The LSD-Fisher test was employed to establish minimum significant differences with a significance threshold of *p* < 0.05. The correlation between variables associated with phenolic compounds, flavonoids, carotenoids, and antioxidant capacity, analyzed by colorimetric methods, was assessed using the Pearson correlation statistic. Additionally, a principal component analysis (PCA) was performed to explore and explain the nature of the variance among the variables using INFOSTAT statistical software version 2020p.

Three independent assays were carried out in triplicate for the human red blood cell hemolysis assay and the SW480 ATCC cell viability assay. A single-factor analysis of variance (ANOVA) was used to determine if differences existed among the groups: treated, positive, and negative controls. The Shapiro–Wilk test was conducted to assess the normality of the data. Tukey’s multiple comparison tests were utilized to identify groups between which statistically significant differences occurred compared to the controls. These tests were performed using the GraphPad Prism v.10 statistical package (San Diego, CA, USA). All tests were statistically significant when *p* < 0.05 with a 95% confidence interval (CI).

## 5. Conclusions

The extracts of the peel of M. flexuosa from Florencia and the seeds of *T. grandiflora* from Versalles, *T. grandiflora* from Balcanes, and *T. grandiflora* from Caraño, revealed antitumor effects against SW480 ATCC colon cancer lines, reducing cell viability by approximately 50% at concentrations of 0.78 to 6.25 µg/mL, curiously at concentrations that did not affect erythrocytes, according to hemolytic tests. In this study, the environmental conditions surrounding the cultivation of *T. grandiflorum* seeds and *M. flexuosa* peel play a crucial role in determining their antioxidant activity. The fruits of *T. grandiflorum* from the Balcanes experimental farm exhibited significantly higher levels of bioactive compounds in the seeds compared to other locations, highlighting the influence of geographical and environmental factors on plant metabolism.

## Figures and Tables

**Figure 1 molecules-30-01250-f001:**
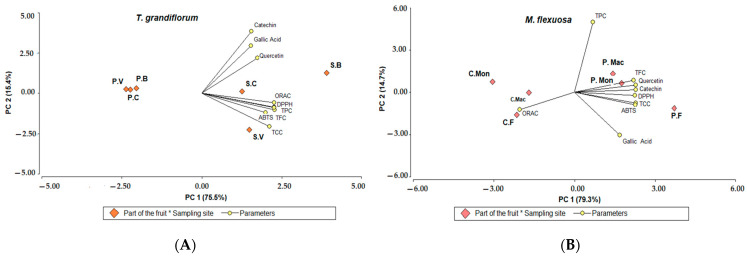
Principal Component Analysis reveals differences in antioxidant metabolites among *T. grandiflorum* seeds and *M. flexuosa* pulp at the different harvest sites. (**A**) PCA from *T. grandiflorum* and (**B**) PCA for *M. flexuosa*. * Interaction Part of the fruit and Sampling site.

**Figure 2 molecules-30-01250-f002:**
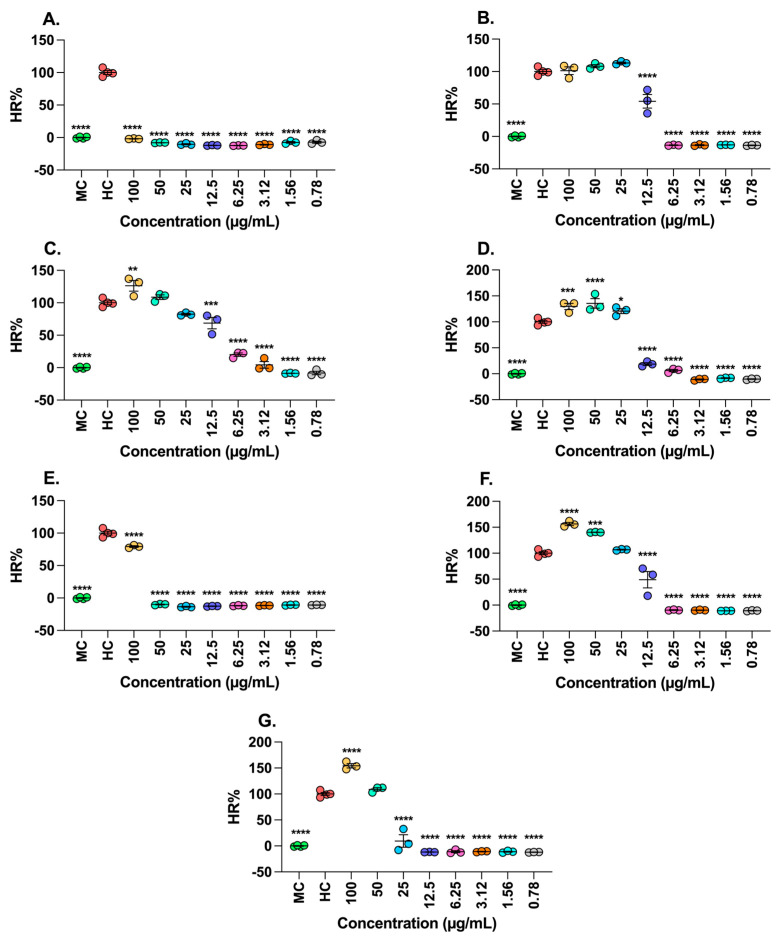
Ex-Vivo Human Red Blood Cell Hemolysis Assay. (**A**) Peel of *M. flexuosa* from Florencia. (**B**) Pulp *M. flexuosa* from Florencia. (**C**) Peel *M. flexuosa* from montañita. (**D**) Pulp of *M. flexuosa* from montañita. (**E**) Seed *T. grandiflora* from *Versalles.* (**F**) Seed *T. grandiflora* from Balcanes. (**G**) Seed *T. grandiflora* from Caraño. Data are depicted as mean ± SEM from three independent assays conducted in triplicate. Results were compared to the Hemolysis Control using Tukey’s multiple comparisons tests. Significance levels are indicated as follows: *p* < 0.05 *, *p* < 0.001 **, *p* < 0.0001 ***, *p* < 0.00001 ****. MC: Medium Control, HC: Hemolysis Control.

**Figure 3 molecules-30-01250-f003:**
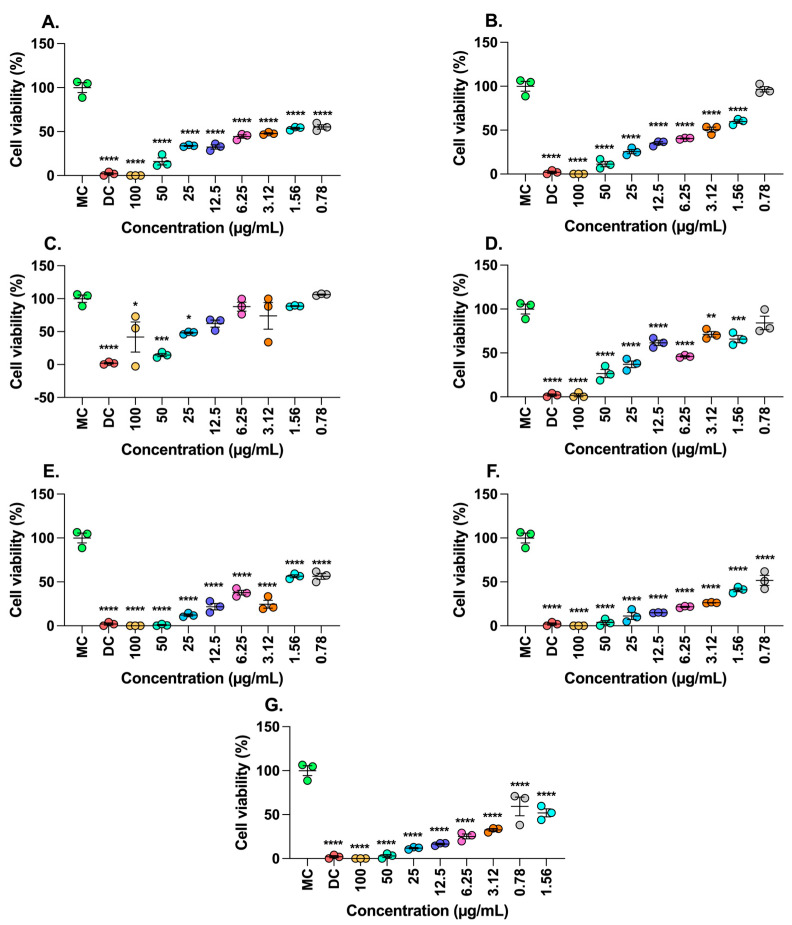
SW480 ATCC Cell viability Assay. Effect of *T. grandiflorum* and *M. flexuosa* fruits on the cell viability of SW480 ATCC cells. (**A**) Peel of *M. flexuosa* Florencia. (**B**) Pulp of *M. flexuosa* Florencia. (**C**) Peel of *M. flexuosa* montañita. (**D**) Pulp of *M. flexuosa* montañita. (**E**) Seed of *T. grandiflora* Versalles. (**F**) Seed of *T. grandiflora* Balcanes. (**G**) Seed of *T. grandiflora* Caraño. Data are depicted as mean ± SEM from three independent assays conducted in triplicate. Results were compared to the medium control using Tukey’s multiple comparisons tests. Significance levels are indicated as follows: *p* < 0.05 *, *p* < 0.001 **, *p* < 0.0001 ***, *p* < 0.00001 ****. MC: Medium Control, DC: Death Control.

**Table 1 molecules-30-01250-t001:** Phenolic compounds contents of *T. grandiflorum* and *M. flexuosa.*

Fruto	F. P *	Place	TPC	TFC	TCC	Gallic Acid	Catechin	Quercetin
*T. grandiflorum*	Seed	Balcanes	619.41 ± 12.05 ^a^	569.09 ± 4.51 ^a^	25.12 ± 0.16 ^b^	335.12 ± 10.87 ^a^	39.86 ± 0.29 ^a^	70.11 ± 0.48 ^b^
		Versalles	524.25 ± 12.05 ^b^	502.85 ± 4.51 ^b^	30.25 ± 0.16 ^a^	26.81 ± 10.87 ^bc^	10.72 ± 0.29 ^e^	6.70 ± 0.48 ^c^
		Caraño	361.17 ± 12.05 ^c^	404.68 ± 4.51 ^c^	22.78 ± 0.16 ^c^	17.92 ± 10.87 ^c^	26.42 ± 0.29 ^b^	79.27 ± 0.48 ^a^
	Pulp	Balcanes	164.21 ± 12.0.5 ^d^	146.87 ± 4.51 ^d^	4.39 ± 0.16 ^d^	51.63 ± 10.87 ^b^	19.07 ± 0.29 ^c^	6.36 ± 0.48 ^cd^
		Versalles	154.12 ± 12.05 ^d^	142.59 ± 4.51 ^d^	1.61 ± 0.16 ^f^	47.31 ± 10.87 ^bc^	16.37 ± 0.29 ^d^	5.01 ± 0.48 ^d^
		Caraño	170.58 ± 12.05 ^d^	141.91 ± 4.51 ^d^	3.03 ± 0.16 ^e^	46.60 ± 10.87 ^bc^	10.72 ± 0.29 ^d^	6.70 ± 0.48 ^d^
*M. flexuosa*	Pulp	Macagual	508.48 ± 10.06 ^a^	206.83 ± 3.92 ^b^	34.92 ± 0.28 ^c^	180.62 ± 0.68 ^d^	38.63 ± 0.56 ^c^	75.20 ± 2.80 ^b^
		Montañita	475.55 ± 10.06 ^b^	206.89 ± 3.92 ^b^	37.24 ± 0.28 ^b^	195.80 ± 0.68 ^c^	45.76 ± 0.56 ^b^	76.00 ± 2.80 ^b^
		Florencia	285.75 ± 10.06 ^d^	223.21 ± 3.92 ^a^	48.00 ± 0.28 ^a^	219.54 ± 0.68 ^a^	61.15 ± 0.56 ^a^	95.50 ± 2.80 ^a^
	Peel	Macagual	354.00 ± 10.06 ^c^	177.23 ± 3.92 ^c^	25.12 ± 0.28 ^d^	175.72 ± 0.68 ^e^	5.68 ± 0.56 ^de^	17.06 ± 2.80 ^c^
		Montañita	367.55 ± 10.06 ^c^	183.00 ± 3.92 ^c^	18.21 ± 0.28 ^f^	125.43 ± 0.68 ^f^	4.27 ± 0.56 ^e^	12.17 ± 2.80 ^c^
		Florencia	217.21 ± 10.06 ^e^	172.79 ± 3.92 ^c^	24.24 ± 0.28 ^e^	197.90 ± 0.68 ^b^	6.43 ± 0.56 ^d^	19.25 ± 2.80 ^c^

* Part of the fruit. TPC, total phenolic content (mg GA/100 g); TFC, total flavonoid content (mg CT/100 g); TCC, total carotenoid content (mg β-Cat/100 g), gallic acid (mg/100 g); catechin (mg/100 g); quercetin (mg/100 g). Data are expressed as mean + EE of 3 independent assays in triplicate. Means with a common letter are not significantly different LSD Fisher; (*p* < 0.05).

**Table 2 molecules-30-01250-t002:** Antioxidant Activity of *T. grandiflorum* and *M. flexuosa*.

Fruit	F. P *	Place	ABTS ^1^	DPPH ^1^	ORAC ^1^
*T. grandiflorum*	Seed	Balcanes	207.63 ± 1.64 ^a^	117.0 ± 1.24 ^a^	204.93 ± 0.41 ^a^
		Versalles	192.05 ± 1.64 ^b^	98.3 ± 1.24 ^b^	170.86 ± 0.41 ^c^
		Caraño	102.19 ± 1.64 ^c^	94.56 ± 1.24 ^b^	185.99 ± 0.41 ^b^
	Pulp	Balcanes	96.89 ± 1.64 ^d^	4.11 ± 1.24 ^c^	67.97 ± 0.41 ^d^
		Versalles	91.34 ± 1.64 ^e^	4.09 ± 1.24 ^c^	50.22 ± 0.41 ^f^
		Caraño	90.66 ± 1.64 ^e^	4.10 ± 1.24 ^c^	58.83 ± 0.41 ^e^
*M. flexuosa*	Pulp	Macagual	52.62 ± 0.65 ^c^	40.40 ± 0.59 ^c^	86.14 ± 2.08 ^f^
		Montañita	59.19 ± 0.65 ^b^	44.28 ± 0.59 ^b^	195.48 ± 2.08 ^d^
		Florencia	71.99 ± 0.65 ^a^	67.23 ± 0.59 ^a^	114.19 ± 2.08 ^e^
	Peel	Macagual	41.98 ± 0.65 ^d^	15.78 ± 0.59 ^d^	256.49 ± 2.08 ^c^
		Montañita	30.17 ± 0.65 ^e^	17.37 ± 0.59 ^d^	526.23 ± 2.08 ^a^
		Florencia	40.06 ± 0.65 ^d^	15.03 ± 0.59 ^d^	471.35 ± 2.08 ^b^

^1^: µmol Trolox.g^−1^ per gram of fresh fruit. * Fruit Portion. Data are expressed as mean + EE of 3 independent assays in triplicate. Means with a common letter are not significantly different LSD Fisher; (*p* < 0.05).

## Data Availability

The data presented in this study are available to the corresponding author upon request. The data are not publicly available due to specific ethical and privacy considerations.

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
