# Peer review of "Phenolic Composition, Antioxidant, and Anti-Proliferative Activities Against Human Colorectal Cancer Cells of Amazonian Fruits Copoazú (*Theobroma grandiflorum*) and Buriti (*Mauritia flexuosa*)"

_molecules, 2025, doi:10.3390/molecules30061250_

Round 1

Reviewer 1 Report

Comments and Suggestions for Authors

In this manuscript, the authors aimed to evaluate the total phenolic content (TFC), total flavonoid content (TFC), total carotenoid content (TCC), content of three phenolic constituents, free radical scavenging capacity, and hemolytic and antiproliferative effects of the hydro-methanolic extracts of the Amazonian fruits, namely Theobroma grandiflorum and Mauritia flexuosa, mainly in vitro. Several samples as seed/pulp or pulp/peel parts from different origins were compared and further analyzed using PCA analysis. Most of the data were properly collected and the results were well discussed. Please address the issues raised by this reviewer prior to publication.

Abstract section

1.     P1, line 31. Please describe the origin of the SW480 cell line.

Introduction section

2.     P2, lines 68-83. If these plants have been used as traditional herbal medicine, please describe.

Results section

3.     P3-P7. Please include proper figure numbers in the text and figure legends. There is inconsistency in the current version. In addition, please indicate Figure 1A or Figure 1B in the text, if you are explaining your results in detail.  

4.     P3 line 96. “… is depicted in Figure 1” should be changed to “… are summarized in Table 1”.

5.     P3, line 100. Typo. FTL should be TFC. bCT should be TCC. (see, line 104)

6.     P3, line 102. Remove a” after “exhibited”.

7.     P3, line 103. Check the data from TFC. “206.83±1.54” should be “223.21±1.18”.

8.     P3-4. Table 1 and Table 2. Please use “,” and “.” in the numbers accurately. Please include statistical results in the figures (e.g., use superscript). The current version did not show any visual information on the data.

9.     P3-4. Table 2. Please spell the name “T. grandiflorum” correctly.

10.  P4, line 130. Don’t you have to mention the ORAC data (see Table 2) in the text?

11.  P5, (current) Figure 3. Please define (A) and (B) in the figures. Please use the defined parameters (not FLT (TPC?), FT (TFC?), carotenoids (TCC?), etc. Some names are missing in the current version (P.V, TPC and TFC (left figure), Quercetin (right figure) etc.).

12.  P5, line 158. “(PC1: 0.71 and PC2: 0.15)” must be “(PC1: 0.79 and PC2: 0.13)”. See figure.

13.  P5, line 163. “(CP1)” should be “(PC1)”.

14.  P5, line 167. (P > 0.05) OK?

15.  P6, line 181. “… that the Peel of M. flexuosa from Florencia” (see current Figure 5B) should read “… that the Seed of T. grandiflora from Versalles” (see current Figure 5E). This reviewer suggests that the names of the parameters may be appeared in the individual figures. Also, please refer to (Figure 1A), (Figure 1B)… one by one in the text so as not to confuse the reader.

16.  P7, (current) Figure 5. Same as above.

Discussion section

17.  P8, line 216. This sentence is written for seeds. Please explain.

18.  P8, line 217. Please note that CIMAZ means the place Macagual.

19.  P8, line 219. “compounds” should be “compounds in pulp”

20.  P9, lines 301 and 306. Typo. “HTC116” should be “HCT116”

Materials and Methods section

21.  P11, line 363. Remove a duplicate word. ”minutes”

22.  P11, line 369. Please describe the origin of the SW480 cell line.

23.  P12, line 424. What does “1.178/5.000” mean?

24.  P13, lines 478-487. The authors used the blood to collect erythrocytes for hemolytic assays. Was this blood collected from human volunteers or from animals? In either case, the authors must provide proper ethical disclosure for conducting experiments on humans or animals.

25.  P13, lines 478-487. And P6-7, (current) Figure 4 and Figure 5. How did the authors perform experiments with positive controls, which are described in the figure legends as HC (hemolysis control) and DC (death control)?

Comments on the Quality of English Language

fare. please check typos as commented.

Author Response

Dear Reviewer, 1,

Thank you very much for taking the time to review this manuscript. Please find the corrections in the re-submitted files.

Reviewer’s Evaluation:

  1. P1, line 31. Please describe the origin of the SW480 cell line.

Response: The cells were obtained by ATCC-USA.

  1. P2, lines 68-83. If these plants have been used as traditional herbal medicine, please describe.

Response: This new paragraph was included with its respective bibliographic reference:

Therefore, in traditional medicine, ziora-anema from the Colombian Amazon has been positioned as a non-pharmacological adjuvant tool in reducing diabetic complications triggered by nitrosative stress and inflammation in the kidneys (16).

  1. P3-P7. Please include proper figure numbers in the text and figure legends. There is inconsistency in the current version. In addition, please indicate Figure 1A or Figure 1B in the text, if you are explaining your results in detail.

Response:

Adjustments were made to the document

  1. P3 line 96. “… is depicted in Figure 1” should be changed to “… are summarized in Table 1”.

Response:

Adjustments were made to the document

  1. P3, line 100. Typo. FTL should be TFC. bCT should be TCC. (see, line 104)

Response:

Adjustments were made to the document

  1. P3, line 102. Remove “a” after “exhibited”.

Response:

Adjustments were made to the document

  1. P3, line 103. Check the data from TFC. “206.83±1.54” should be “223.21±1.18”.

Response:

Adjustments were made to the document

  1. P3-4. Table 1 and Table 2. Please use “,” and “.” in the numbers accurately. Please include statistical results in the figures (e.g., use superscript). The current version did not show any visual information on the data.

Response:

The number of tables in the text and in their titles were adjusted, the data of the tables had an adjustment to express the results with ±EE and significant differences according to LSD-Fisher p<0.05. Besides, the statistical analysis with superscript (*) was adjusted in Figures 2 and 3.

  1. P3-4. Table 2. Please spell the name “T. grandiflorum” correctly.

Response:

Adjustments were made to the document

  1. P4, line 130. Don’t you have to mention the ORAC data (see Table 2) in the text?

Response: They are described in lines 116-120.

  1. P5, (current) Figure 3. Please define (A) and (B) in the figures. Please use the defined parameters (not FLT (TPC?), FT (TFC?), carotenoids (TCC?), etc. Some names are missing in the current version (P.V, TPC and TFC (left figure), Quercetin (right figure) etc.).

Response:

Figures 1A and 1B were renamed by adjusting the parameter labels.

P5, line 158. “(PC1: 0.71 and PC2: 0.15)” must be “(PC1: 0.79 and PC2: 0.13)”. See figure.

The values ​​were adjusted to those reported in the new figures 1A and 1B

  1. P5, line 163. “(CP1)” should be “(PC1)”.

Response:

Adjustments were made to the document

  1. P5, line 167. (P > 0.05) OK?

Response:

It was written due to an editing error; the correlation analysis has no significance; therefore, it was removed in this version.

  1. P6, line 181. “… that the Peel of M. flexuosa from Florencia” (see current Figure 5B) should read “… that the Seed of T. grandiflora from Versalles” (see current Figure 5E). This reviewer suggests that the names of the parameters may appear in the individual figures. Also, please refer to (Figure 1A), (Figure 1B) … one by one in the text so as not to confuse the reader.

Response:

Adjustments were made to the figures.

  1. P7, (current) Figure 5. Same as above.

Response:

Adjustments were made to the figures.

  1. P8, line 216. This sentence is written for seeds. Please explain.

Response:

It was presented this way due to an editing error. It was already adjusted by adding seeds, which presented the highest levels, see table 1.

  1. P8, line 217. Please note that CIMAZ means the place Macagual.
  2.  

Response:

Adjustments were made to the document.

  1. P8, line 219. “compounds” should be “compounds in pulp”

Response:

It was presented this way due to an editing error. It was already adjusted by adding “pulp” term, which presented the highest levels, see table 1.

  1. P9, lines 301 and 306. Typo. “HTC116” should be “HCT116”

Response:

Adjustments were made to the document.

  1. P11, line 363. Remove a duplicate word. ”minutes”

Response:

Adjustments were made to the document

  1. P11, line 369. Please describe the origin of the SW480 cell line.

Response:

ATCC-USA Origin.

  1. P12, line 424. What does “1.178/5.000” mean?

Response:

It was a typing error. The adjustment to the document was made.

  1. P13, lines 478-487. The authors used the blood to collect erythrocytes for hemolytic assays. Was this blood collected from human volunteers or from animals? In either case, the authors must provide proper ethical disclosure for conducting experiments on humans or animals.

Response:

The blood collection was carried out under the informed consent of a human donor, for which the consent was previously signed; this form was delivered to the Molecules Journal.

  1. P13, lines 478-487. And P6-7, (current) Figure 4 and Figure 5. How did the authors perform experiments with positive controls, which are described in the figure legends as HC (hemolysis control) and DC (death control)?

Response:

Adjustments were made to the graphs.

Kind regards,

Manuscript ID molecules-3114532

(On behalf of all coauthors)

Reviewer 2 Report

Comments and Suggestions for Authors

1. It is better to use the full name for abbreviations that appear for the first time in the abstract.

2. In the abstract, line 24, is there a missing TPC data?

3. In Table 1 and 2, the decimal points of data are used in confusion.

4. Does the age of fruit trees affect the content of fruit ingredients? Did the authors take this into account when collecting fruit samples?

5. You may wish to consider having your paper professionally edited for English language. There are multiple errors in the manuscript, for example, line 363 ‘for 30 minutes. minutes and was …’, line 371 ‘was reached., was preserved…’, the incomplete bracket in line 406. Please check the full text and revise.

6. The purity and supplier for all standards, the supplier and grade of all chemical reagents, the manufacture of all equipment, and the source of any statistical or other software programs must be specified.

7. Line 389, 421, and 479: There must be a space between the number and the unit symbol, please check the entire manuscript and modify it.

8. Line 424 and 494: What does the number mean?

9. Lines 430: No space is required between ℃ and numbers, please unify the use of this unit throughout manuscript.

10. Lines 417, 422, 443, and 482, please note the use of superscripts and subscripts throughout the manuscript.

11. Line 136 ‘Figure 5a’ and 170 ‘Figure 5b’, textual content does not match the picture.

12. The image encoding starts from Figure 3? In figure 3, the coordinate heading is CP1 and CP2? Figure 3a, the data point near P.C and P.B are not labeled.

13. Line 139-148, these data information can not be observed in Figure 3.

14. The study is relatively simple with slightly less data, is it possible to performed a detailed compositional analysis of the metabolites of these samples?

Comments on the Quality of English Language

You may wish to consider having your paper professionally edited for English language.

Author Response

August 8th, 2024

Dear Reviewer 2,

Thank you very much for taking the time to review this manuscript. Please find the corrections in the re-submitted files.

Reviewer’s Evaluation:

  1. It is better to use the full name for abbreviations that appear for the first time in the abstract.

Response:

The full name and abbreviations were adjusted in the abstract.

  1. In the abstract, line 24, is there missing TPC data?

Response:

The missing data was added.

  1. In Table 1 and 2, the decimal points of data are used in confusion.

Response:

In Tables 1 and 2, decimal points were corrected.

  1. Does the age of fruit trees affect the content of fruit ingredients? Did the authors take this into account when collecting fruit samples?

Response:

No, tree age was not a study variable in this research. However, the influence of growing space was evaluated.

  1. You may wish to consider having your paper professionally edited for English language. There are multiple errors in the manuscript, for example, line 363 ‘for 30 minutes. minutes and was …’, line 371 ‘was reached., was preserved…’, the incomplete bracket in line 406. Please check the full text and revise.

Response:

Grammatical adjustments were made to the indicated lines.

  1. The purity and supplier for all standards, the supplier and grade of all chemical reagents, the manufacture of all equipment, and the source of any statistical or other software programs must be specified

Response:

The adjustments were made to the manuscript.

  1. Line 389, 421, and 479: There must be a space between the number and the unit symbol, please check the entire manuscript and modify it.

Response:

The spaces between the number and the unit symbol were corrected along the paper.

  1. Line 424 and 494: What does the number mean?

Response:

The numbers were corrected.

  1. Lines 430: No space is required between ℃ and numbers, please unify the use of this unit throughout manuscript.

Response:

The spaces between ℃ and numbers were corrected

  1. Lines 417, 422, 443, and 482, please note the use of superscripts and subscripts throughout the manuscript.

Response:

The adjustments were made to the manuscript as required.

  1. Line 136 ‘Figure 5a’ and 170 ‘Figure 5b’, textual content does not match the picture. 

Response:

The adjustments were made to the manuscript

  1. The image encoding starts from Figure 3. In figure 3, the coordinate heading is CP1 and CP2?. Figure 3a, the data point near P.C and P.B are not labeled.

Response:

The adjustments were made to the manuscript.

  1. Line 139-148, this data information cannot be observed in Figure 3.

Response:

The adjustments were made to the manuscript.

  1. The study is relatively simple with slightly less data, is it possible to perform a detailed compositional analysis of the metabolites of these samples?

Response:

We appreciate your comment. However, for this study the isolation of compounds was not contemplated, considering the complexity of the total composition of the samples derived from Amazonian fruits and the possible synergy between the substances. However, the analysis by high-performance liquid chromatography was carried out, which allowed us to demonstrate the presence of compounds such as gallic acid, quercetin and catechin. In the future, we hope to have more funding that will allow us to isolate and purify some compounds and test their individual bioactivity.

Kind regards,

Manuscript ID molecules-3114532

(On behalf of all coauthors)

Reviewer 3 Report

Comments and Suggestions for Authors

The aim of the study is not clear. From the description of their results and conclusion, the authors emphasized how the results varied with the sampling location, which suggests a focus on the influence of geographical location and climatic conditions. However, these factors were not discussed in the introduction or adequately discussed with reference to the literature.  The methodology is not clear and lacks important information. Below are additional comments.

1.      What do the authors mean by the fruits were collected in the department of Caquetá, Colombia? The statement should be revised.

2.      The authors should provide the climatic conditions including the annual rainfall, temperature and humidity for the areas where the fruits were collected.

3.      The differences between section 4.2 and section 4.3 are not clear. According to their study what is the difference between samples collect and fruit collection?

4.      The authors should explain how the samples were packed/handled during transportation, treated before analysis, the pulp and seed were separated?

5.      How did the authors prepare the pulp and seed samples?

6.      What do the authors mean by Capturing capacity of the ABTS radical equivalent of trolox (TEAC ABTS·+) and Trolox equivalent DPPH radical capturing capacity (TEAC DPPH). The authors should try to use common terminology found in the literature to avoid ambiguity.

7.      Which software was used with the generalized linear and mixed model to determine the comparison between the response variables?

8.      The statistics is not clear? For instance, the ** p < 0.001 can not be linked with Table 1. 

Comments on the Quality of English Language

The manuscript requires extensive editing of the English.

Author Response

August 8th, 2024

Dear Reviewer 3,

Thank you very much for taking the time to review this manuscript. Please find the corrections in the re-submitted files.

Reviewer’s Evaluation:

  1. What do the authors mean by the fruits were collected in the department of Caquetá, Colombia? The statement should be revised.

Response:

The distribution in Colombia is by regions, departments, and municipalities. However, to make it easier to understand, we have modified the text in a general way.

  1. The authors should provide the climatic conditions including the annual rainfall, temperature and humidity for the areas where the fruits were collected.

Response:

Climate information was added in section 4.2.

  1. The differences between section 4.2 and section 4.3 are not clear. According to their study what is the difference between sample collect and fruit collection?

Response:

The sections 4.2 and 4.3 were consolidated into one section. The section 4.2  was titled "Sampling and fruit collection area"

  1. The authors should explain how the samples were packed/handled during transportation, treated before analysis, the pulp and seed were separated?

Response:

The following passage was added at the end of section 4.2 “During the post-harvest, the fruits were transported to the laboratory under refrigeration. In addition, manual extraction was necessary to separate the pulp and peel (M. flexuosa), and to separate pulp and seeds (T. grandiflorum).”

  1. How did the authors prepare the pulp and seed samples?

Response:

This information can be found in section 4.3.

  1. What do the authors mean by Capturing capacity of the ABTS radical equivalent of trolox (TEAC ABTS·+) and Trolox equivalent DPPH radical capturing capacity (TEAC DPPH). The authors should try to use common terminology found in the literature to avoid ambiguity

Response:

In the new version of the manuscript, terminology was adjusted according to the terms commonly used in the literature.

  1. Which software was used with the generalized linear and mixed model to determine the comparison between the response variables?

Response:

All statistical analyzes were carried out using INFOSTAT version 2020p, which is specified in the methods session in lines 522-531.

  1. The statistics is not clear? For instance, the ** p < 0.001can not be linked with Table 1. 

Response:

An adjustment was made in the statistical analysis expressing the results as mean + EE of 3 independent assays in triplicate. Means with a common letter are not significantly different LSD Fisher; p < 0.05). The changes can be seen in tables 1 and 2.

Kind regards,

Manuscript ID molecules-3114532

(On behalf of all coauthors)

Round 2

Reviewer 1 Report

Comments and Suggestions for Authors

Thank you for your revision. In the current version, not all but most of the corrections are suitable for publication. Before publishing, please make sure whether Informed Consent Statement is NOT APPLICABLE is right (see line 570), because you have returned as below. Please state it accordingly.

#24 P13, lines 478-487. The authors used the blood to collect erythrocytes for hemolytic assays. Was this blood collected from human volunteers or from animals? In either case, the authors must provide proper ethical disclosure for conducting experiments on humans or animals.

Response:

The blood collection was carried out under the informed consent of a human donor, for which the consent was previously signed; this form was delivered to the Molecules Journal.

Author Response

  •  The manuscript title doesn’t fit the content of the manuscript. I suggest the following
    “Biological properties and anti-proliferative activity of Amazonian fruits copoazú
    (Theobroma grandiflorum) and Buriti (Mauritia flexuosa) extracts against human
    colorectal cancer cells”.

Response: Adjustments were made to the title. However, the analyzed phenolic composition
we consider is of equal relevance. 

  • Abstract, line 32, thesentence “ … the number of viable cells in the SW480 ATCC
    cell line …” must be updated as following “… the number of viable human colorectal
    cancer cells, using SW480 ATCC cell line, …”

Response: The corrections in the manuscript were made.

  • Introduction, Lines 94-96. The aim of the work must be described more accurately.
    It is not “ … identify the content of secondary metabolites with radical scavenging
    capacity and antioxidant activity.” The authors identified the secondary metabolites
    responsible for the biological properties???? What are they?? Gallic acid, Catechin
    and Quercetin?? By sure they will be much more!!!

Response: In accordance with their recommendations, the research objective was changed. 

  • The full stop (.) is not necessary in the titles of the different sections and subsections
    The subheadings of section 2 are not adequate since they seems more related with
    results highlights than a title. For instance: “2.1. Theobroma grandiflorum seeds and
    Mauritia flexuosa pulp showed higher bioactive compound concentrations than other
    fruit fractions.” “2.2. T. grandiflorum seeds and M. flexuosa peel exhibit antioxidant
    activity” “2.3. Principal Component Analysis reveals differences in antioxidant
    metabolites among fruit 137 parts and the harvest sites” “2.4. In vitro assays show
    that extracts of de T. grandiflorum and M. flexuose cause hemolysis at high 
    concentrations” Doesn’t correspond to subheadings. Please correct all of them 
    accordingly. The same observation for the captions of Tables 1and 2, needs to be
    adjusted. Section 4 must be re-structured since it is a little confusing. 4.4.
    Quantification of phenolic compounds by (HPLC-UV) The title of the subheading “4.4. Determination of bioactive compounds” must be changed as follow “ 4.5.
    Evaluation of biological properties” Within this subheading the sequence should be:
    4.5.1. Total phenolic content (TPC) 4.5.2. Total flavonoid content (TFC) 4.5.3. Total
    carotenoid content (TCC) 4.6. Antioxidant activity 4.6.1. Determination of Total
    Antioxidant Capacity by Oxygen Radical Absorption 446 (ORAC) 4.6.2. ABTS
    Assay 4.6.3. DPPH Radical Scavenging Capacity 4.7. Hemolytic activity 4.8.
    Antiproliferative activity 4.9. Determination of cell viability by incorporating crystal
    violet 4.10. Statistical analysis. 

Response: The manuscript's section and subsection headings were revised to omit full stop.
Furthermore, the titles of individual items were reorganized. However, we determined that
combining biological activity  with overall metabolite determination is inappropriate.
Because, they are different sections of compound quantification and determination of
biological activity.

  • Line 402 “… Gallic acid (0 - 550 ppm, R2: 0.9986) was used as a standard.” Must be
    updated as following “ Gallic acid was used as reference standard for quantification
    purposes in a concentration range from XX (lowest concentration level) to 500 ppm.”
    The lowest concentration can’t be “Zero” ppm, of course. Please correct the value
    accordingly.

Response: The lowest concentration level was corrected, in each calibration curve.

Furthermore, please find attached the Change of Authorship Form. After a careful review of
each author's contributions, from the initial research idea to the data analysis and discussion,
the research team has decided to revise the author order. We apologize for any oversight in
not addressing this matter in our previous submission.
